# Impact of COVID-19 pandemic on acute stroke admissions and case-fatality rate in lower-income and middle-income countries: a protocol for systematic review and meta-analysis

Martin Ackah ,[1,2] Mohammed Gazali Salifu ,[2,3] Louise Ameyaw,[2] Hosea Boakye,[4] Cynthia Osei Yeboah[1]

[1]Department of Physiotherapy, Korle Bu Teaching Hospital, Accra, Ghana
[2]School of Public Health, Department of Epidemiology and Disease Control, University of Ghana, Legon, Greater Accra, Ghana
[3]Monitoring and Evaluation Directorate, Ministry of Health, Accra, Ghana
[4]School of Public Health, Department of Biological, Environmental, and Occupational Health, University of Ghana, Legon, Ghana

**Correspondence to**
Martin Ackah;
martinackah10@gmail.com

## ABSTRACT

**Introduction** The current review primarily aims to ascertain the impact of COVID-19 on stroke admission as well as stroke case fatality in Low-income and Middle-Income Countries (LMICs).

**Methods and analysis** Four international databases (PubMed/Medline, Google Scholar, African Journals Online, Latin American and Caribbean Health Sciences Literature) and one preprint database (medRxiv). Studies will be included if they are conducted in LMICs, all stroke types without age and language restriction, from December 2019 to 31 December 2021. Two authors will screen the titles and abstracts against the prespecified eligibility criteria for inclusion in the review, and then repeat the process after retrieving the full text. Joanna Briggs critical appraisal checklist for analytical cross-sectional studies will be used for the quality assessment and risk of bias by two coauthors. The characteristics of the studies will be presented and summarised in a table. We aim to perform meta-analyses on a pooled proportional change in acute stroke admissions and case fatality with 95% CI using a random-effects meta-analysis. Publication bias will be assessed using funnel plot and Egger's regression test if ≥10 studies are involved. A sub group analysis will be performed to determine the sources of heterogeneity. Leave-one-out sensitivity analysis will be performed to examine the impact of a single study on the overall pool estimate.

**Ethics and dissemination** Ethical approval is not required as this is secondary research and will use reported data in scientific literature. A full manuscript will be submitted to a reputable peer-review journal for publication and disseminated electronically and in print.

**PROSPERO registration number** CRD42021281580.

## Strengths and limitations of this study

► A well-validated systematic review and meta-analysis methods that are fully compliant with established international norms and guidelines would be followed.
► To examine the robustness of the derived estimates, a sensitivity and subgroup analysis would be done.
► There would be no language restriction.
► As a result of the regional and cultural variances, there would be significant heterogeneity across the studies.

worldwide as of 25 September 2021, with Africa accounting for approximately 3% of global incidence.[2]

COVID-19 is largely a respiratory infection, but research in patients with severe infections has showed that it is a multi-system inflammatory condition with neurological effects such as stroke.[3] For example, stroke rates among confirmed and hospitalised COVID-19 patients have been found to range from 1.4% to 1.74% according to recent global systematic review and meta-analyses.[4 5] In 2021, the global predicted case-fatality rate (CFR) of COVID-19 ranged from 10%[6] to nearly 18%.[7]

There have been conflicting accounts on the impact of Acute Stroke Admission (ASA) appearing in emergency rooms. For example, a systematic review and meta-analysis showed that the proportion of haemorrhagic stroke admissions increased in the pandemic period compared with the proportion of haemorrhagic stroke admission in the prepandemic era by 10%.[8] The study was limited by several factors; first, the total number of admissions during the pandemic was not reported, rendering the incidence of stroke inconclusive; second, the lack of data about stroke

## INTRODUCTION

The WHO announced a SARS Cov-2 pandemic in March 2020, posing a threat to healthcare systems and cultures around the world.[1] Since the first recorded case in December of 2019, more than 240 million COVID-19 cases and 4.7 million deaths have been confirmed

**BMJ**

severity, ischaemic stroke sub types, and prognoses, might have underestimated/overestimated the pandemic's impact on stroke to some extent.[8] Conversely, a global systematic review and meta-analysis on the impact of COVID-19 pandemic on ASA found a significant reduction of 29% compared with prepandemic period.[9] The authors did not conduct a subgroup analysis to determine the influence of the pandemic on ASA in Low-income and Middle-Income Countries (LMICs). Furthermore, just a few papers from LMICs were included in the prior review. Obviously, this cannot be generalised to represent the panoramic view of COVID-19 impact of ASA in LMICs.

Despite documented high-quality services with respect to workflow metrics, angiographic results, complications, outcomes at discharge, maintenance of reperfusion therapies as well as infection control measures,[10 11] the pandemic has had a negative impact on stroke CFR in both high income countries and LMICs based on observational studies.[10 12 13] For instance, stroke CFR increased by 2.4 percentage points in the UK,[10] and by a significant 60% in France.[12] A cross-sectional study in sub-Saharan Africa found an estimated peripandemic high stroke CFR of 29.3%.[13]

Several primary investigations have been carried out to determine the peri-pandemic stroke CFR in LMICs with widely varying reported magnitudes. The authors are unaware of any previous systematic reviews and/or meta-analyses on the subject in LMICs. With this in mind, the authors sort to conduct a systematic review and meta-analysis on the impact of COVID-19 pandemic on acute stroke hospitalisation and death in limited-resource countries in the world.

## RESEARCH QUESTIONS
1. What is the impact of COVID-19 pandemic on ASA in LMICs?
2. What is the impact of the COVID-19 pandemic on post-stroke CFR in LMICs?

## METHOD AND ANALYSIS
### Best practices and registration
The current protocol was reported in compliance with Preferred Reporting Items for Systematic Reviews and Meta-analyses Protocol (PRISMA-P)[14] (checklist file). The study will start on 30 April 2022 and end on 30 August 2022. A final paper will be published in a reputable peer-review journal.

### Eligibility criteria
#### Studies
Observational studies such as longitudinal, cohort, case–control and cross-sectional studies reporting the impact of COVID-19 pandemic on stroke admission and case fatality in LMICs will be considered in the current review. Case series, case reports, narrative and systematic reviews, editorials, comments, letters, opinion pieces, abstracts,

conference proceedings and animal studies will all be excluded.

### Participants
We aim to include impact of COVID-19 pandemic on ASA and CFR studies involving all age group from LMICs. All stroke types (ischaemic vs haemorrhagic stroke vs transient ischaemic stroke) will be included. In-patient stroke participants in all care structure concern such as hospital, and health centres in LMICs will be included. General medical admission and studies from HIC will be excluded. LMICs are countries defined by World Bank Group.[15]

### Intervention/exposure
Exposure is the COVID-19 pandemic.

### Comparators
The study will compare rates of stroke admissions and case fatality between pre-COVID-19 pandemic era and peri/post-COVD-19 pandemic era.

### Outcome of interest
The outcome of interest will be: (1) the impact of COVID-19 pandemic on ASA in LMICs and (2) the impact of COVID-19 pandemic on stroke CFR in LMICs.

### Data source and search strategies
Four international databases (ie, PubMed/Medline, Google Scholar, African Journals Online, Latin American and Caribbean Health Sciences Literature] and one preprint database (medRxiv) will be searched for relevant studies on the impact of COVID-19 pandemic on stroke admissions and/or stroke case fatality in LMICs from December 2019 to December 2021 without language restriction.

Medical Subject Headings (MeSH) terms and free text will be used in the search approach. These terms will be combined with the Boolean operators 'OR' and 'AND'. The key term includes; COVID-19, Stroke, cerebrovascular accident, admissions, case fatality, low-income and middle-income countries, Africa, sub-Saharan Africa, Asia, Latin America, Caribbeans. The full search string is shown in online supplemental file 1.

### Other sources
The authors will evaluate the reference lists of papers that meet the inclusion and exclusion criteria to see whether there will be any additional studies that are luckily to be missed by our electronic search.

### Data screening and selection
Two independent authors will independently review the titles and abstracts against the pre-specified eligibility criteria for inclusion in the review, and then repeat the process after retrieving the full text. A disagreement will be resolved by discussion or reference to a third reviewer.

### Data extraction and management
The data extraction will be done by two researchers independently using a pretested and standardised excel

spreadsheet.[16] Discrepancies arising during abstract and full text review will be discussed and agreement reached by adjudication of a third author. Data such as the author's name, year of publication, country, study design, patients' characteristics (ie, sex, age), prepandemic stroke admission and case fatality, peripandemic stroke admission and case fatality, total sample size will be extracted. Mendeley will be used to managed and remove duplicated articles.

### Addressing missing data

If any of the selected articles contain inadequate information, we will contact the appropriate author through email to request missing data. If that is not possible, the data will be deleted. Furthermore, the authors will obtain raw data from the primary study authors wherever possible so that we can extract the missing data.

### Quality assessment and risk of bias

Joanna Briggs critical appraisal checklist for analytical cross-sectional studies will be used for the quality assessment and risk of bias.[17] The checklist's purpose is to assess the methodological quality of each study that will be included in the review. In systematic reviews, it is one of the most extensively used appraisal tools. In addition, the instrument was chosen because of its objectivity and ease of use. The tools consist of eight questions with the following answers; yes, no, unclear and not applicable with correct and rigorous methodology assigned yes responses. The quality assessments and bias risk assessments will be carried out by two independent peer reviewers. Discrepancies discovered during the abstract and full text reviews will be explored and resolved through discussion. To reduce selection bias, reviewers will be blinded to the article's author, journal and year of publication during the title and abstract review.

### Data synthesis

Extracted data will be exported into Stata (V.16; Stata) from Microsoft excel 2013 for all analyses. The study selection procedure will be summarised using the 2020 PRISMA flow chart[18] (figure 1). The characteristics of the studies will be presented and summarised in a table. We aim to perform meta-analyses on pooled proportional change in ASA and CFR with 95% CI using a random-effects model. With regard to case fatality, inpatient, 7 days, 14 days ad 30 days CFR will be pooled where possible. The heterogeneity of effect size estimates across these studies will be quantified using the $I^2$ statistic and Q statistic's p value.[19] Publication bias will be assessed using funnel plot and Egger's regression test if ≥10 studies are involved.

Where significant heterogeneity exists in the included studies, a subgroup analysis will be performed to determine the sources of heterogeneity based on the following, economic status (low-income country and middle-income country), continent (Africa, South America, Asia), stroke type (ischaemic vs haemorrhagic), study type (prospective

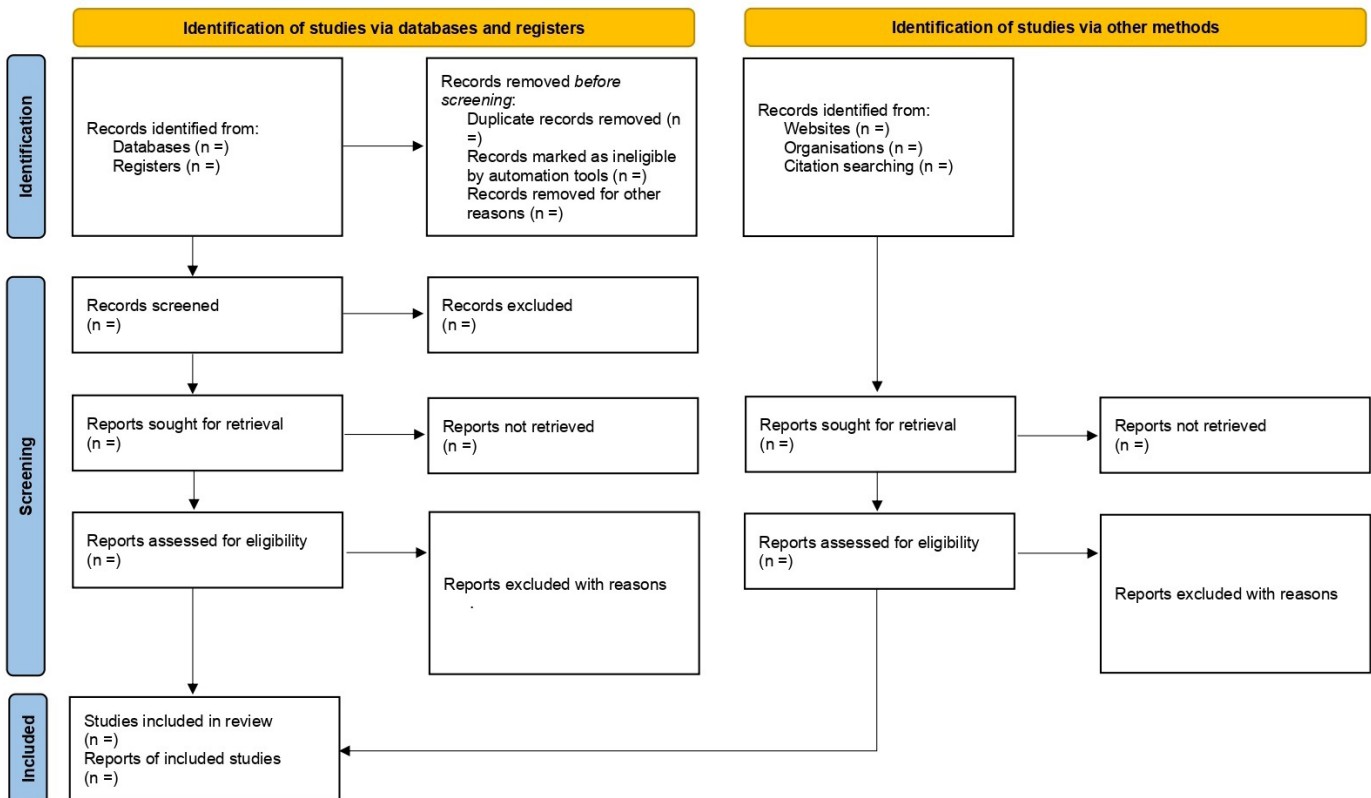

**Figure 1** PRISMA 2020 flow diagram for new systematic reviews which included searches of databases, registers and other sources. PRISMA, Preferred Reporting Items for Systematic Reviews and Meta-Analyses.

vs retrospective, national/regional vs single hospital) and sample size (≤400 vs >400). Leave-one-out sensitivity analysis will be performed to examine the impact of a single study on the overall pooled estimate.

## Patients and public involvement

Patient or the public were not involved in the design, or conduct, or reporting, or dissemination plans of our research.

## ETHICS AND DISSEMINATION

Ethical approval is not required as this is secondary research and will use reported data in scientific literature. A full manuscript will be submitted to a reputable peer-review journal for publication.

**Contributors** MA conceived the study, drafted the manuscript, critically revised the manuscript for methodological and intellectual content. MGS drafted the manuscript, critically revised the manuscript for methodological and intellectual content. LA critically revised the manuscript for methodological and intellectual content. HB drafted the manuscript, critically revised the manuscript for methodological and intellectual content. COY drafted the manuscript, critically revised the manuscript for methodological and intellectual content. All authors approved the final manuscript. MA is the guarantor of the review.

**Funding** The authors have not declared a specific grant for this research from any funding agency in the public, commercial or not-for-profit sectors.

**Competing interests** None declared.

**Patient and public involvement** Patients and/or the public were not involved in the design, or conduct, or reporting, or dissemination plans of this research.

**Patient consent for publication** Not applicable.

**Provenance and peer review** Not commissioned; externally peer reviewed.

**ORCID iDs**
Martin Ackah http://orcid.org/0000-0002-5045-1104
Mohammed Gazali Salifu http://orcid.org/0000-0003-2025-5337

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
