## [Reviewer comments · BMJ Open]

ARTICLE DETAILS

TITLE (PROVISIONAL)	Impact of COVID-19 pandemic on acute stroke admissions and case-fatality rate in Lower- and Middle-Income Countries: a protocol for systematic review and meta-analysis.
AUTHORS	Ackah, Martin; Salifu, Mohammed Gazali; Ameyaw, Louise; Boakye, Hosea; Yeboah, Cynthia Osei

VERSION 1 – REVIEW

REVIEWER	Chan, Bernard National University Health System, Medicine
REVIEW RETURNED	28-Oct-2021

GENERAL COMMENTS	This protocol for a systematic review on the impact of the COVID-19 pandemic on acute stroke admission and case-fatality in lower and middle-income countries address an important topic on stroke care. However, both stroke admission rates and mortality rates have been affected by several co-variables during the COVID-19 epidemic, which have already been published quite extensively. These co-variables include, but are not limited to, stroke subtypes (ischaemic vs haemorrhagic stroke, and whether TIA admissions were included in ischaemic stroke), stroke severity (NIHSS score on admission, and whether in-patient strokes that complicated severe COVID-19 pneumonia were included), stroke without or with concomitant COVID-19 pneumonia. Other factors that are likely important include the incidences of COVID-19 during the pandemic in a particular country/region, and the characteristics of the included studies (e.g. prospective vs retrospective, national/regional vs single hospital or sample size). As the above-mentioned factors could affect both presentation to hospitals of stroke patients and the severity of stroke patients hospitalised during the COVID-10 pandemic or included in different studies, it is important to (1) Define the stroke patients to be included and the periods of pre-, peri- and post-pandemic more strictly, and (2) Include some of these factors in the sub-group analyses and define them a priori. Careful sub-group analyses could be hypothesis generating for any change in the stroke hospitalisation and mortality during the COVID-19 pandemic.
--

REVIEWER	Lesaine, Emilie CHU de Bordeaux
REVIEW RETURNED	20-Jan-2022

GENERAL COMMENTS	This article presents the study protocol of a systematic review and meta-analysis that aims to investigate the impact of Coronavirus
--

	Disease-19 (COVID-19) on stroke admission and case fatality in low- and middle-income countries (LMICs). Stroke is an emergency condition that requires rapid and specific management and represents one of the main causes of disability and death in many countries. The issue of the impact of the COVID pandemic on stroke admissions and mortality is of great interest, because it has potentially important implications on the management of stroke patients, particularly in LMICs, where few studies have been conducted. However, the article suffers from many limitations, particularly regarding the quality of the justification and the consistency between the objectives and the method.  • Introduction The authors present the heterogeneity of results in the literature concerning the evolution of stroke admissions during the Covid crisis as well as that of mortality during this phase. However, no hypotheses are presented regarding the mechanisms explaining these variations. In particular, the authors do not present any hypothesis explaining these variations in developing countries that would justify the conduct of their study. The only justification presented is the small number of studies on the subject carried out in developing countries. The authors have planned to measure the impact of the Covid crisis on stroke admissions and post-stroke mortality using a systematic review and meta-analysis. However, this method does not fit to the objectives. At most, the authors will be able to analyze the evolution of stroke admissions and post-stroke mortality during the Covid crisis. In the discussion, a reflection concerning the Covid crisis impact on post-stroke admissions and mortality, i.e. in terms of a causal relationship between the crisis and the results, could possibly be carried out, according to the results found and taking into account the quality and the methods applied in the selected studies. Moreover, the presentation of the referenced studies results is very imprecise (non-specification of COVID crisis periods, mortality times, study populations when comparing countries) and leads to errors of interpretation.  Method The authors are intended to include all studies on the impact of COVID-19 pandemic on stroke admission and case fatality from LMICs, without selection criteria on age. The inclusion criteria are not precise enough: what are the analysis periods, types of stroke included, are patients with COVID infection included ? Are pediatric strokes really included, on what rationale? For the objective on the analysis of the impact of COVID-19 on stroke admissions; in particular it would be useful to specify the admission services (emergency departments only, inpatient department) and care structures concerned (hospitals, health centers). For the analysis of the impact of COVID-19 on post-stroke mortality, it is necessary to know the time frame for mortality (in-hospital, 7 days, 30 days...) and the causes of death selected (all causes, cardio-neuro-vascular).  • Expected key results and discussion This paragraph does not provide any discussion on the expected results, strengths, limitations and implications of this study.
--	--

REVIEWER	Caminiti, Caterina University Hospital of Parma, Research and Innovation Unit
REVIEW RETURNED	24-Jan-2022

GENERAL COMMENTS	This is the protocol of a review concerning a very current and relevant research topic. A main strength of this work is that it is focused on Lower- and Middle-Income Countries, and that it is conducted by researchers of this area. Also, 4 databases are searched, and the methodology appears to be rigorous. Please find my suggestions below, framed according to the PRISMA guidelines. Title I suggest rephrasing the title as follows: “Impact of COVID-19 pandemic on acute stroke admissions and case-fatality in Lower- and Middle-Income Countries: a protocol for systematic review and meta-analysis” Abstract The abstract should define study and population eligibility criteria, briefly describe the selection process, and give information on the tool used to assess risk of bias (quality). Introduction It is a bit fuzzy. A set of data is provided on the trends in various countries drawn from a selection of small observational studies which do not necessarily represent the trend in that country. I suggest referring to existing systematic reviews only (the one by Reddy et al already cited, and others such as by You et al. doi: 10.1136/bmjopen-2021-050559; Katsanos et al. doi: 10.1161/STROKEAHA.121.034601). Also, when indicating trends, the periods of references should be given. All statements should be supported by literature. In the following sentence: “Despite documented high-quality services, the pandemic has had a negative impact on stroke case fatality in both HIC and LMICs based on observational studies”, both claims regarding high quality of services and impact of the pandemic on fatality should be justified. Methods In the “Quality assessment and risk of bias” paragraph, the choice for the instrument should be justified, and the tool briefly described. Discussion This section offers the opportunity to describe the possible implication for research and practice of the review, which would be interesting to convey the importance of this work.
---

VERSION 1 – AUTHOR RESPONSE

Reviewer: 1

comment: This protocol for a systematic review on the impact of the COVID-19 pandemic on acute stroke admission and case-fatality in lower and middle-income countries address an important topic on stroke care. However, both stroke admission rates and mortality rates have been affected by several co-variates during the COVID-19 epidemic, which have already been published quite extensively. These co-variates include, but are not limited to, stroke subtypes (ischaemic vs haemorrhagic stroke, and whether TIA admissions were included in ischaemic stroke), stroke severity (NIHSS score on admission, and whether in-patient strokes that complicated severe COVID-19

pneumonia were included), stroke without or with concomitant COVID-19 pneumonia. Other factors that are likely important include the incidences of COVID-19 during the pandemic in a particular country/region, and the characteristics of the included studies (e.g. prospective vs retrospective, national/regional vs single hospital or sample size).

As the above-mentioned factors could affect both presentation to hospitals of stroke patients and the severity of stroke patients hospitalised during the COVID-19 pandemic or included in different studies, it is important to (1) Define the stroke patients to be included and the periods of pre-, peri- and post-pandemic more strictly, and (2) Include some of these factors in the sub-group analyses and define them a priori. Careful sub-group analyses could be hypothesis generating for any change in the stroke hospitalization and mortality during the COVID-19 pandemic.

Authors response: The authors have taken the reviewer's comment into account and have defined the stroke patients to be included (page 6) as well as the pandemic date/study period are now clearly established (page 7). In addition, sub-group analysis of some possible factors that might influence the objectives of the study have now been included (Page 10).

Reviewer: 2

This article presents the study protocol of a systematic review and meta-analysis that aims to investigate the impact of Coronavirus Disease-19 (COVID-19) on stroke admission and case fatality in low- and middle-income countries (LMICs).

Stroke is an emergency condition that requires rapid and specific management and represents one of the main causes of disability and death in many countries. The issue of the impact of the COVID pandemic on stroke admissions and mortality is of great interest, because it has potentially important implications on the management of stroke patients, particularly in LMICs, where few studies have been conducted.

However, the article suffers from many limitations, particularly regarding the quality of the justification and the consistency between the objectives and the method.

- Introduction

comment: The authors present the heterogeneity of results in the literature concerning the evolution of stroke admissions during the Covid crisis as well as that of mortality during this phase. However, no hypotheses are presented regarding the mechanisms explaining these variations. In particular, the authors do not present any hypothesis explaining these variations in developing countries that would justify the conduct of their study. The only justification presented is the small number of studies on the subject carried out in developing countries. The authors have planned to measure the impact of the Covid crisis on stroke admissions and post-stroke mortality using a systematic review and meta-analysis. However, this method does not fit to the objectives. At most, the authors will be able to analyze the evolution of stroke admissions and post-stroke mortality during the Covid crisis. In the discussion, a reflection concerning the Covid crisis impact on post-stroke admissions and mortality, i.e. in terms of a causal relationship between the crisis and the results, could possibly be carried out, according to the results found and taking into account the quality and the methods applied in the selected studies.

Moreover, the presentation of the referenced studies results is very imprecise (non-specification of COVID crisis periods, mortality times, study populations when comparing countries) and leads to errors of interpretation.

Response: The authors thank the reviewer for this observation. First the individuals/observational studies which produced heterogenous outcomes have now been removed and replaced with systematic reviews only. Second, the limitations in the individual reviews have been highlighted which gives credence to the conduct of this current review. With respect to the discussion, the section is currently removed as recommended by the editor.

Method

comment: The authors are intended to include all studies on the impact of COVID-19 pandemic on stroke admission and case fatality from LMICs, without selection criteria on age. The inclusion criteria are not precise enough: what are the analysis periods, types of strokes included, are patients with COVID infection included? Are pediatric strokes really included, on what rationale?

For the objective on the analysis of the impact of COVID-19 on stroke admissions; in particular it would be useful to specify the admission services (emergency departments only, inpatient department) and care structures concerned (hospitals, health centers). For the analysis of the impact of COVID-19 on post-stroke mortality, it is necessary to know the time frame for mortality (in-hospital, 7 days, 30 days...) and the causes of death selected (all causes, cardio-neuro-vascular).

Response: The authors thank the reviewer for the comment. First of all, the review plan to include all age groups as stated the under the participants' subsection. Second, the analysis period will commence from December, 2019 to December, 2021, as mentioned under data sources and search strategies subsection. Third, all stroke types such as ischemic vs haemorrhagic stroke vs transient ischemic stroke] will be included as stated under the participants' section. Moreover, in-patient stroke participants in all care structure concern such as hospital, and health centers in LMICs will be included. With regards to case fatality, inpatient, 7days, 14 days, ad 30 days will be pooled where possible as well as the cause of death.

comment: Expected key results and discussion

This paragraph does not provide any discussion on the expected results, strengths, limitations and implications of this study.

Response: This section is removed as recommended by the editor

Reviewer: 3

This is the protocol of a review concerning a very current and relevant research topic. A main strength of this work is that it is focused on Lower- and Middle-Income Countries, and that it is conducted by researchers of this area. Also, 4 databases are searched, and the methodology appears to be rigorous.

Please find my suggestions below, framed according to the PRISMA guidelines.

comment: I suggest rephrasing the title as follows:

"Impact of COVID-19 pandemic on acute stroke admissions and case-fatality in Lower- and Middle-Income Countries: a protocol for systematic review and meta-analysis"

Response: Thanks for the comment. Authors have now rephrased the title as suggested/recommended.

comment: The abstract should define study and population eligibility criteria, briefly describe the selection process, and give information on the tool used to assess risk of bias (quality).

Response: Thanks for the comment. The suggested information are now included in the abstract. The statements read " Studies will be included if they are conducted in LMICs, all stroke types without age and language restriction, from December, 2019 to 31 December, 2021. Two authors will screen the titles and abstracts against the pre-specified eligibility criteria for inclusion in the review, and then repeat the process after retrieving the full-text. Joanna Briggs critical appraisal checklist for analytical cross-sectional studies will be used for the quality assessment and risk of bias by two co-authors.

Introduction

Comment: It is a bit fuzzy. A set of data is provided on the trends in various countries drawn from a selection of small observational studies which do not necessarily represent the trend in that country. I suggest referring to existing systematic reviews only (the one by Reddy et al already cited, and others such as by You et al. doi: 10.1136/bmjopen-2021-050559; Katsanos et al. doi:

10.1161/STROKEAHA.121.034601). Also, when indicating trends, the periods of references should be given.

Response: Authors have taken the reviewer's comment into consideration, and accordingly removed the observational studies, and replaced with existing systematic reviews only as recommended (highlighted in page 4-5). It now reads " There have been conflicting accounts on the impact of Acute Stroke Admission (ASA) appearing in emergency rooms. For example, a systematic review and meta-analysis showed that the proportion of haemorrhagic stroke admissions increased in the pandemic period compared to the proportion of haemorrhagic stroke admission in the pre-pandemic era by 10%⁸. The study was limited by several factors; first, the total number of admissions during the pandemic was not reported, rendering the incidence of stroke inconclusive; second, the lack of data about stroke severity, ischemic stroke sub types, and prognoses, might have underestimated/overestimated the pandemic's impact on stroke to some extent⁸. Conversely, a global systematic review and meta-analysis on the impact of COVID-19 pandemic on ASA found a significant reduction of 29% compared to pre-pandemic period⁹. The authors did not conduct a subgroup analysis to determine the influence of the pandemic on ASA in LMICs. Furthermore, just a few papers from LMICs were included in the prior review. Obviously, this cannot be generalized to represent the panoramic COVID-19 impact of acute ASA in LMICs."

Comment: All statements should be supported by literature. In the following sentence: "Despite documented high-quality services, the pandemic has had a negative impact on stroke case fatality in both HIC and LMICs based on observational studies", both claims regarding high quality of services and impact of the pandemic on fatality should be justified.

Response: Thanks for the comment. As recommended, all statements are now supported and justified by literature. The paragraph now reads " Despite documented high-quality services with respect to workflow metrics, angiographic results, complications, outcomes at discharge, maintenance of reperfusion therapies as well as infection control measures^{10,11}, the pandemic has had a negative impact on stroke CFR in both high income countries and LMICs based on observational studies^{12–14}"

Methods

comment: In the "Quality assessment and risk of bias" paragraph, the choice for the instrument should be justified, and the tool briefly described.

Responses: The authors will like to thank the reviewer for the comment. The choice of the appraisal tool is now justified and briefly explained. The paragraph reads " Joanna Briggs critical appraisal checklist for analytical cross-sectional studies will be used for the quality assessment and risk of bias¹⁸. The checklist's purpose is to assess the methodological quality of each study that will be included in the review. In systematic reviews, it is one of the most extensively utilized appraisal tools. In addition, the instrument was chosen because of its objectivity and ease of use. The tools consist of eight questions with the following answers; yes, no, unclear, and not applicable with correct and rigorous methodology assigned yes responses"

Discussion

Comment: This section offers the opportunity to describe the possible implication for research and practice of the review, which would be interesting to convey the importance of this work.

Response: This section is removed as recommended by the editor